# Wind Loading on Scaled Down Fractal Tree Models of Major Urban Tree Species in Singapore

**Woei-Leong Chan [1],\*, Yong Eng [1],\*, Zhengwei Ge [2], Chi Wan Calvin Lim [2], Like Gobeawan [2], Hee Joo Poh [2], Daniel Joseph Wise [2], Daniel C. Burcham [3], Daryl Lee [3], Yongdong Cui [1] and Boo Cheong Khoo [1]**

[1] Department of Mechanical Engineering, National University of Singapore, 9 Engineering Drive 1, Block EA #07-08, Singapore 117575, Singapore; cui_yongdong@nus.edu.sg (Y.C.); mpekbc@nus.edu.sg (B.C.K.)

[2] Fluid Dynamics Department, Institute of High Performance Computing, Agency for Science, Technology and Research (A*STAR), 1 Fusionopolis Way, #16-16 Connexis, Singapore 138632, Singapore; ge_zhengwei@ihpc.a-star.edu.sg (Z.G.); limcw@ihpc.a-star.edu.sg (C.W.C.L.); gobeawanl@ihpc.a-star.edu.sg (L.G.); pohhj@ihpc.a-star.edu.sg (H.J.P.); daniel-wise@ihpc.a-star.edu.sg (D.J.W.)

[3] Centre for Urban Greenery and Ecology (CUGE), National Parks Board, Singapore Botanic Gardens, 1 Cluny Road, Singapore 259569, Singapore; Daniel_Christopher_BURCHAM@nparks.gov.sg (D.C.B.); Daryl_LEE@nparks.gov.sg (D.L.)

\* Correspondence: woeileong.chan@nus.edu.sg (W.-L.C.); e0023457@u.nus.edu (Y.E.)

**Abstract:** Estimation of the aerodynamic load on trees is essential for urban tree management to mitigate the risk of tree failure. To assess that in a cost-effective way, scaled down tree models and numerical simulations were utilized. Scaled down tree models reduce the cost of experimental studies and allow the studies to be conducted in a controlled environment, namely in a wind or water tunnel, but the major challenge is to construct a tree model that resembles the real tree. We constructed 3D-printed scaled down fractal tree models of major urban tree species in Singapore using procedural modelling, based on species-specific growth processes and field statistical data gathered through laser scanning of real trees. The tree crowns were modelled to match the optical porosity of real trees. We developed a methodology to model the tree crowns using porous volumes filled with randomized tetrahedral elements. The wind loads acting on the tree models were then measured in the wind tunnel and the velocity profiles from selected models were captured using particle image velocimetry (PIV). The data was then used for the validation of Large Eddy Simulations (LES), in which the trees were modelled via a discretized momentum sink with 10–20 elements in width, height, and depth, respectively. It is observed that the velocity profiles and drag of the simulations and the wind tunnel tests are in reasonable agreement. We hence established a clear relationship between the measured bulk drag on the tree models in the wind tunnel, and the local drag coefficients of the discretized elements in the simulations. Analysis on the bulk drag coefficient also shows that the effect of complex crown shape could be more dominant compared to the frontal optical porosity.

**Keywords:** fractal tree model; porous volume; wind loading; particle image velocimetry; large eddy simulation; frontal area density; frontal silhouette area; frontal optical porosity

## 1. Introduction

Wind is one of the major causes of tree failure, and it is essential to accurately understand the wind loads affecting individual trees during risk assessment. Numerical simulation offers a cost-effective way to address the challenge.

In simulations, the tree is usually modelled as a momentum sink, but it is difficult to prescribe the drag coefficient correctly [1–10]. One of the most detailed numerical simulations of a single tree was conducted using Reynolds Average Navier–Stokes (RANS) simulation [11]. The study used a discretized momentum sink to simulate the wind drag on a tree, and the drag coefficient was tuned until the simulation outcome matched the experimental measurements. The method, while effective, still requires multiple simulations to determine the proper drag coefficient. The authors also observed that the drag coefficient is highly dependent on how well the tree model is described, and hence it is difficult to compare the drag coefficients of different studies. We applied the same tuning methodology in our earlier study [12]. In this study, a novel definition of local drag coefficient can provide appropriate value without going through the tuning process. The definition establishes a direct connection between the bulk drag of the tree and the local drag coefficient. Hence, making it possible to compare the local drag coefficient, if the bulk drag and total frontal silhouette area of the discretized elements are known.

We chose to conduct the study using Large Eddy Simulations (LES) due to the nature of the highly turbulent flow behind the tree models. LES has been used to analyze the flow above and within vegetation canopies [13–15]. It has been further applied at the street scale to study the aerodynamic impact of vegetation on the urban environment [16].

To validate the simulation results, we conducted experiments using scaled down fractal tree models. Scaled down fractal tree models reduce the cost of experimental investigation on the flow field around large standing trees. It also allows the study to be conducted in a controlled environment, namely within a wind tunnel [17–22] or a water tunnel [23,24]. However, one of the major challenges is to construct tree models that resemble the real trees. We constructed tree models of seven major urban tree species in Singapore based on statistical data gathered through laser scanning of the real trees. The tree species are *Khaya senegalensis* (Senegal mahogany), *Hopea odorata* (chengal pasir), *Samanea saman* (rain tree), *Swietenia macrophylla* (broad-leafed mahogany), *Syzygium grande* (sea apple), *Tabebuia rosea* (trumpet tree), and *Peltophorum pterocarpum* (yellow flame). The *P. pterocarpum* tree model used in this study is more elaborate than the one used in our previous studies [12,25].

Our main aim in this study is to achieve the following goals: (1) devise a wind load simulation methodology that works on trees; (2) validate the simulation methodology using experiments; (3) collect species-specific bulk drag coefficients. Furthermore, the non-flexible tree models and the usage of a wind tunnel reduces uncertainties in the experiments so that the simulations can be properly validated. The tests also provide reference bulk drag coefficients important for practical use for a quick tree failure assessment in the field, especially for trees with weakened trunks that are more vulnerable to low wind speeds.

## 2. Materials and Methods

### 2.1. Tree Models

The photos of the tree models are as shown in Figure 1. The tree models were fabricated using Selective Laser Sintering (SLS) 3D-printing. The material of choice was Nylon 3200 Glass Filled or Nylon PA12, depending on the printable volume. The models were all built at 200 mm in height ($H$). At the height of 200 mm, the Reynolds number of the wind tunnel tests is significantly lower than the Reynolds number of the real trees, but the eventual bulk drag coefficients measured in the wind tunnel are within reasonable range compared to real trees at low wind speed, as shown later in the paper.

The tree model development was divided into two phases: (1) the development of the fractal tree model, which is made up of the trunk and branches; (2) the development of the tree crown.

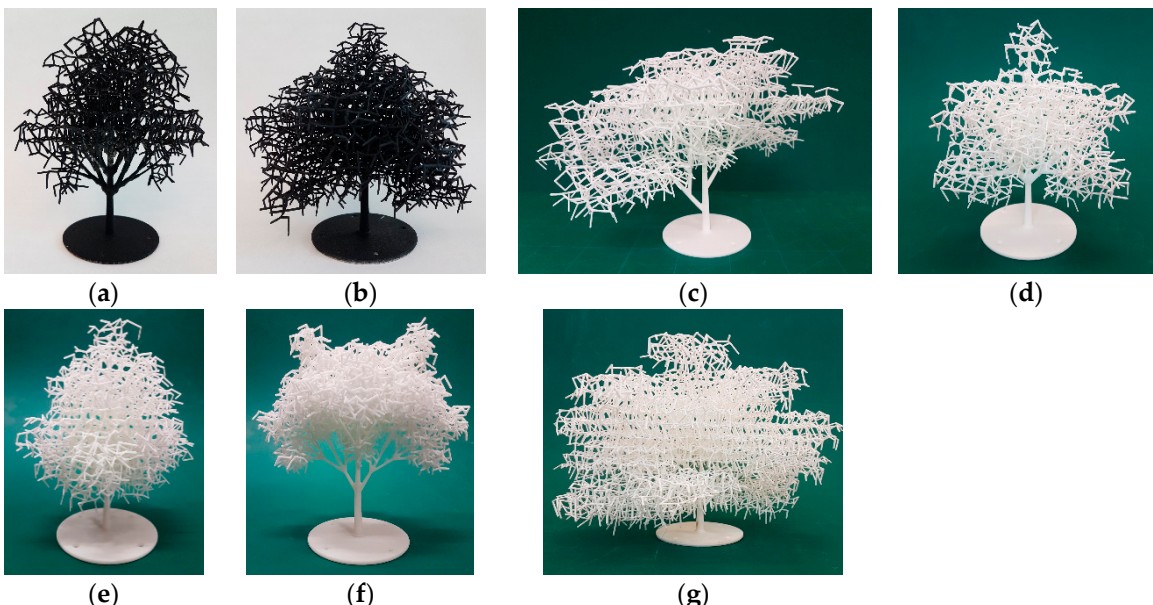

**Figure 1.** Fabricated tree models: (**a**) *Khaya senegalensis*; (**b**) *Hopea odorata*; (**c**) *Samanea saman*; (**d**) *Swietenia macrophylla*; (**e**) *Syzygium grande*; (**f**) *Tabebuia rosea*; (**g**) *Peltophorum pterocarpum*. *K. senegalensis,* and *H. odorata* were painted black to reduce laser reflections in particle image velocimetry (PIV) measurements.

### 2.1.1. Development of the Fractal Tree Models

We generated the fractal tree models using procedural modelling based on species-specific growth processes and statistical data gathered through laser scanning of the real trees. The tree models were produced using pre-formulated L-systems [26,27]. This is a well-established fractal mechanism in botany, used to generate the tree species models based on a set of rules and parameters which mimic the growth and transformation of living cells. The growth rules were constructed specifically to match the tree architecture of a target species, and the rules dictate the tree growth from a seed into different stages. The result of this process is an "average" tree model consisting of the trunk and branches that can represent the target tree species. It is important that this approach is adopted to use average representative species models rather than any individual trees reconstructed directly from laser scanning. A laser scan of an individual tree is often prone to acquisition noise, leaves, or epiphytes, all of which conceal the actual branching structure of the tree. Use of a single scan would therefore result in reconstructed tree models that may not represent the actual branching pattern of a species accurately. Details of the fractal tree model development is described by Gobeawan et al. [28]. In this work, the higher generation branches that were deemed too thin for fabrication were ignored, and the lower generation branches that cannot be ignored were thickened to at least 2 mm in diameter, but the overall shape of the trees were not greatly compromised.

### 2.1.2. Development of the Tree Crown Models

Due to the impracticality of fabricating the scaled down leaves, the tree crowns were represented by a porous volume constructed using randomized tetrahedral elements. Instead of arbitrarily modelling the shape of the crown, we chose to define the crown volume using ellipsoids centered at the tips of the branches such that the resultant crown shape is similar to the target species. Figure 2 shows the process of crown development for the *K. senegalensis* tree model. The interconnecting randomized tetrahedral elements filled the volumes defined by the ellipsoids centered at the tip of the branches. The same processes of other tree models are depicted in Figures A1–A6 in Appendix A.

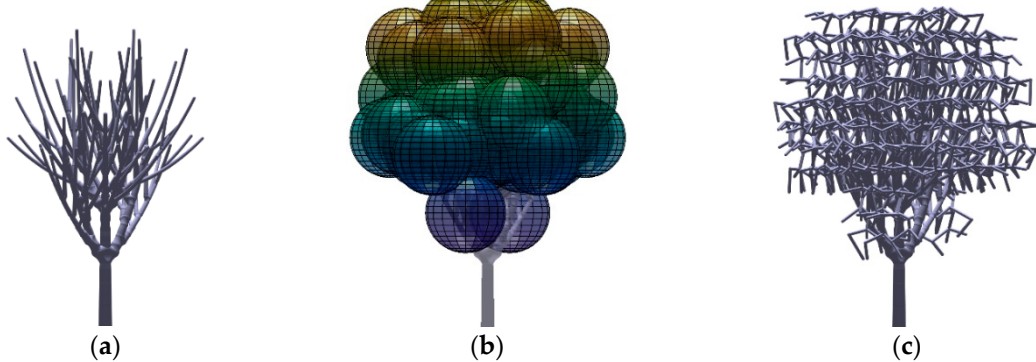

**Figure 2.** *K. senegalensis* fractal tree model: (**a**) trunk and branches; (**b**) tree crown volume; (**c**) final model (frontal view at 0° rotation).

We generated the tree crown to have frontal optical porosity close to the real trees. The frontal optical porosity was calculated after the crown was merged with the trunk and branches. The process was repeated with incremental change of the average dimensions of the tetrahedral elements until the frontal optical porosity of the tree model was as close to the data obtained from the real trees as possible.

To obtain optical porosity of the real tree, photographs of healthy, non-excessively pruned, mature trees with fully developed leaves were taken and converted into binary image to determine the optical porosity of the frontal view. We chose to remove the background manually instead of automatically [29,30] due to the urban setting of our photos. Furthermore, we do not need the optical porosity to be extremely accurate because it is not possible to have exact match between the real tree and the tree model. Only the areas bounded by the crown were taken into account for the determination of optical porosity as shown in Figure 3. The frontal crown area is the area encapsulated by the outer boundary of the tree crown (area occupied by black and white pixels in Figure 3c, and the optical porosity is the area occupied by the void (white pixels) divided by the frontal crown area. Table 1 summarizes the optical porosities of the real trees. The optical porosity is between 0 and 1, of which 0 indicates non-porous.

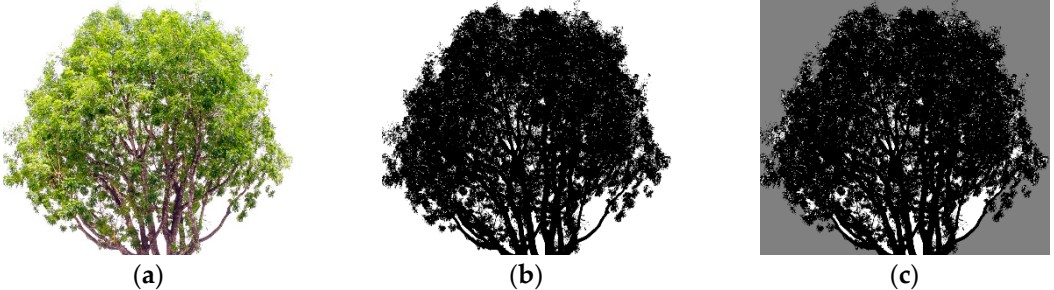

**Figure 3.** *K. senegalensis*: (**a**) photo of real tree; (**b**) binary image; (**c**) image with defined crown area.

**Table 1.** Optical porosities of the real trees.

| Tree Species | Sample Size | Optical Porosity | |
|---|---|---|---|
| | | Average | Standard Deviation |
| *K. senegalensis* | 5 | 0.1237 | 0.0711 |
| *H. odorata* | 5 | 0.1005 | 0.0732 |
| *S. saman* | 7 | 0.1864 | 0.0743 |
| *S. macrophylla* | 11 | 0.1424 | 0.0506 |
| *S. grande* | 7 | 0.1157 | 0.0573 |
| *T. rosea* | 12 | 0.1172 | 0.0421 |
| *P. pterocarpum* | 7 | 0.1583 | 0.0499 |

## 2.2. Wind Tunnel Tests

The developed tree models were then tested in the Temasek Laboratories closed-loop subsonic wind tunnel with a contraction ratio of 12:1, and a square test section of $0.6 \times 0.6 \times 2$ m in width, height, and length. The wind tunnel generates uniform flow with a thin boundary layer near the walls. Figure 4 shows the wind tunnel. The main objective of the wind tunnel tests is to obtain experimental data for the validation of the simulations. The tests were conducted at wind speeds of 5, 10, and 15 m/s.

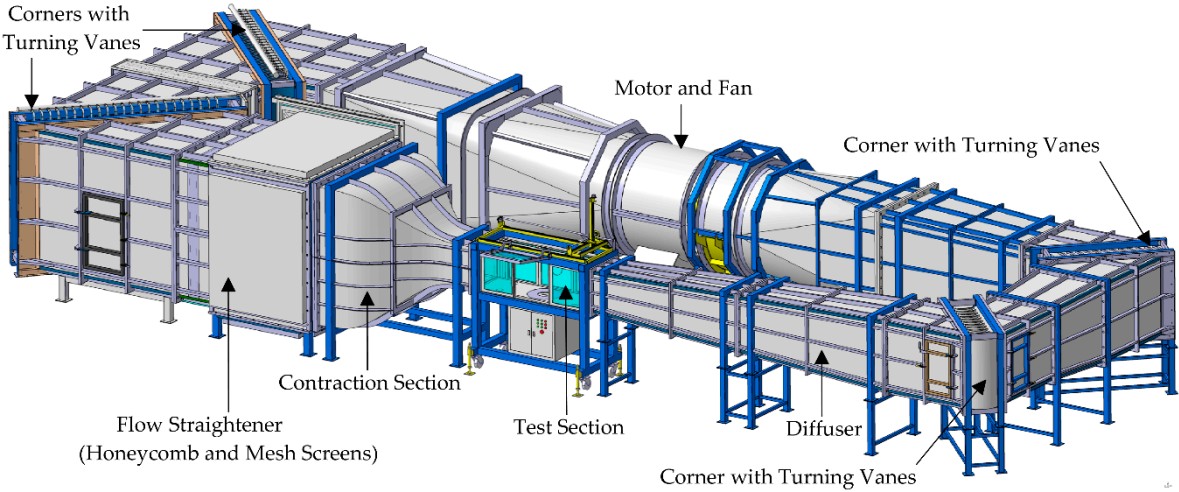

**Figure 4.** Closed-loop subsonic wind tunnel.

### 2.2.1. Wake Profile Measurement

The flow field velocity vectors at the streamwise center plane of the *K. senegalensi*s and *H. odorata* at 0°, 45°, and 90° rotation angles were measured using particle image velocimetry (PIV). The PIV system consists of a Phantom Miro M320s high speed camera from Vision Research, Inc., a laser system from Litron Lasers (an LDY304 PIV laser, a laser guiding arm, and laser sheet optics), and a high-speed controller from LaVision GmBH. PIV data was taken for a duration of 4 s at 300 Hz. The post-processing software is DaVis 8 [31] from Lavision GmBH. The experimental setup is illustrated in Figure 5.

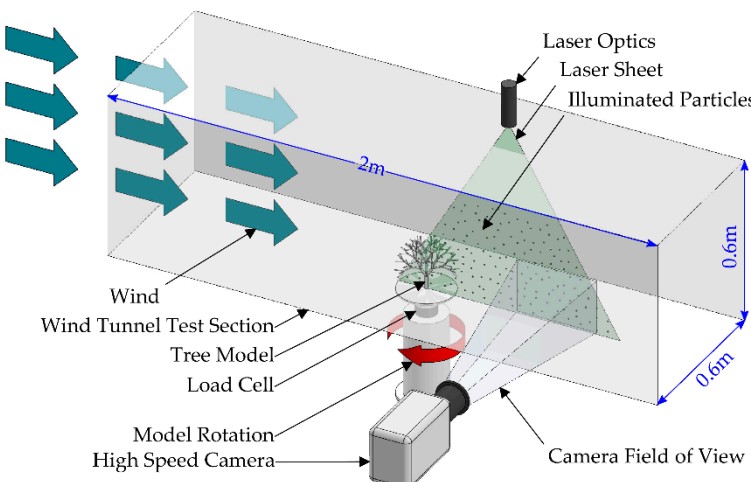

**Figure 5.** Illustration of wind tunnel experimental setup.

### 2.2.2. Bulk Drag Measurement

The bulk drag, which is the total drag of the whole tree model was measured using a load cell mounted at the bottom of the tree. The load cell is the six-component ATI Gamma from ATI Industrial

Automation. Only the drag force was measured in this study. For each measurement, drag force was recorded for a duration of 5 s at 1000 Hz and repeated at least three times. The drag coefficient was calculated using Equation (1).

$$C_D = \frac{2D}{\rho A_{ref} U_{ref}^2}$$
(1)

$A_\text{ref}$ = Reference area;
$C_D$ = Bulk drag coefficient;
$D$ = Bulk drag measurement in wind tunnel;
$U_\text{ref}$ = Reference wind speed;
$\rho$ = Air density.

*2.3. Numerical Simulations*

2.3.1. Solver and Numerical Models

In this study, the LES simulations were performed with OpenFOAM [32], an open-source finite volume code. The incompressible Navier–Stokes equations were solved via the PIMPLE algorithm with the Wall-adapting Local Eddy-viscosity (WALE) subgrid scale model [33]. The WALE model is known to provide adequate results for wall-bounded flows [33–36].

While the tree was not explicitly modelled, the volumetric tree crown was modelled using appropriate momentum sink terms. The simulation code was modified to implement the momentum sink terms in the Navier–Stokes equations as shown in Equations (2)–(4).

$$\frac{\partial u_i}{\partial x_i} = 0$$
(2)

$$\frac{\partial u_i}{\partial t} + \frac{\partial u_i u_j}{\partial x_j} = -\frac{\partial p}{\partial x_i} + \frac{\partial}{\partial x_j}\left[ v_t \left( \frac{\partial u_i}{\partial x_j} + \frac{\partial u_j}{\partial x_i} \right) - \frac{2}{3}\delta_{ij}k \right] + S_{u_i}$$
(3)

$$S_{u_i} = -(C_d \cdot \text{FSAD})u_i U$$
(4)

FSAD = Frontal silhouette area density, frontal silhouette area divided by tree crown volume;
$S_{u_i}$ = Momentum sink;
$C_d$ = Drag coefficient;
$U$ = Velocity magnitude = $\sqrt{u_i u_i}$ (using the Einstein summation convention);
$u_i$ = Velocity component.

2.3.2. Grid and Boundary Conditions

The setup of the study is sketched in Figure 6. The computational domain is a box with dimensions $3H \times 3H \times 13.5H$ in the width, height, and length, where $H$ is the tree height. The generated mesh is finer near the wall. The tree refined volume was defined as 1.5 times the tree's largest dimension in width and height, stretching from $1H$ upstream to $2H$ downstream. In this region, the grid resolution $\delta$ is $H/50$. The mesh was gradually enlarged between the regions which corresponds to a total of 7.1 million number of cells in the domain. The computational work was done using resources of the National Supercomputing Centre (NSCC), Singapore.

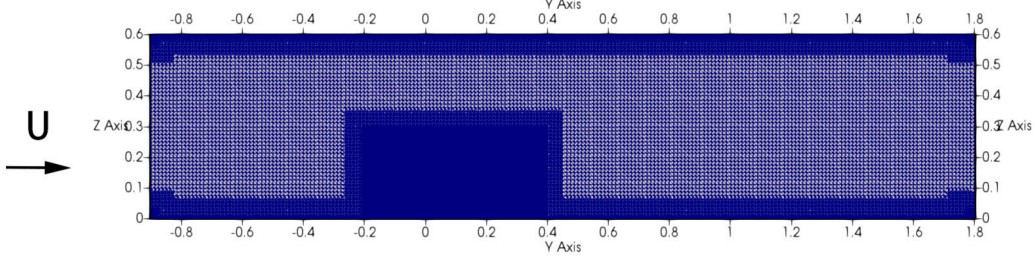

**Figure 6.** Schematic of simulation domain. The dimensions are in meters.

A turbulent inlet boundary condition was used at $4.5H$ upstream. The mean and root mean square of the inlet flow were maintained to be the same as the data obtained from the wind tunnel; consequently, a random fluctuating component was added to the mean field. The outlet boundary is at $9H$ downstream, where a zero gradient assumption was used in the direction normal to the outlet plane. No-slip boundary conditions were applied to the walls. The time step size was limited by keeping the maximum Courant number below 1. This results in an averaged time step of $1.5 \times 10^{-4}$ s. The residual convergence criteria used in this study was $10^{-6}$. It measures the iterative solution's convergence and directly quantifies the error in the solution of the system of equations. The simulations were initialized until the velocity and pressure field data were statistically stable. In the current work, statistical stability is reached when the difference in a time-averaged quantity is less than 0.1% when the averaging period is halved. In this study, the mean drag and velocity profiles at $1H$ and $2H$ are examined to ensure the solutions are statistically stable.

### 2.3.3. Tree Modelling

In this study, the trees were modelled within the simulations as a discretized porous volume. The momentum sink representing the porous effect was discretized in the X (spanwise), Y (streamwise), and Z (height) directions for each tree species. Each of the discretized elements consists of two local parameters: $C_d$ and FSAD. The symbol $C_d$ is different from the bulk drag coefficient $C_D$ in Equation (1), distinguishable by the subscripted capital $D$ in Equation (1) and small letter $d$ in Equation (5). The parameters are defined in Equations (5) and (6):

$$C_d = \frac{2D}{\rho A_{total} U_{ref}{}^2} \tag{5}$$

$$\text{FSAD}_n = \frac{A_{fs_n}}{V_{e_n}} \tag{6}$$

$$A_{total} = \sum_{n=1}^{N} A_{fs_n} \tag{7}$$

$A_{\text{total}}$ = Total frontal optical silhouette area;
$A_{\text{fs}}$ = Frontal optical silhouette area;
$C_d$ = Local drag coefficient;
$D$ = Drag force measurement in wind tunnel;
$N$ = Total number of discretized elements;
$V_e$ = Volume of element.

These parameters depend on both the tree species being tested, and the flow direction. Equations (1) and (5) establish the connection between the bulk drag coefficient of the whole tree and the local drag coefficient for the discretized elements in the simulation. The tree model was discretized into different resolutions: $\Delta = 5^3$, $10^3$, and $20^3$ elements, namely 5, 10, and 20 elements in the directions of height, width, and depth respectively. This resulted in 125, 1000, and 8000 total elements for each respective resolution. The 9th and 20th slices for *K. senegalensis* tree model are presented in Figure 7. In the 9th slice, the branches show up near to the trunk of the tree model. In the final slices, the tree crown branches no longer exist at the outermost point of the tree crown. The frontal silhouette area of each slice is presented in black. The silhouette area is the product of the number of black pixels and the area represented by each pixel. The summation of frontal silhouette area in all slices is the total frontal silhouette area; which was used as the reference area in the calculation of the local drag coefficient. The resultant local $C_d \cdot$ FSAD is presented in Figure 7d. The trunk has the greatest local value. The streamwise averaged and longitudinal averaged coefficient of *K. senegalensis* at different resolutions are plotted in Figure 8. In the tree model volume, the center of the crown has the largest values. Similar plots for other tree models are presented in Figure A7 in Appendix B.

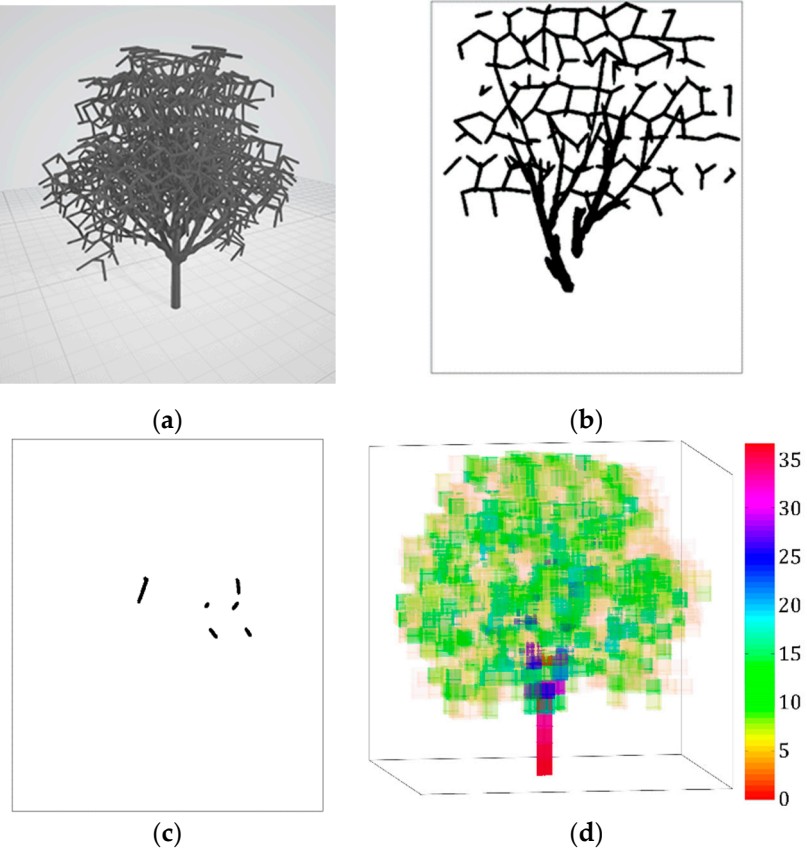

**Figure 7.** (**a**) *K. senegalensis* tree model resolution $\Delta = 20^3$; (**b**) the 9th slice; (**c**) the 20th slice; (**d**) resultant local $C_d \cdot$ FSAD.

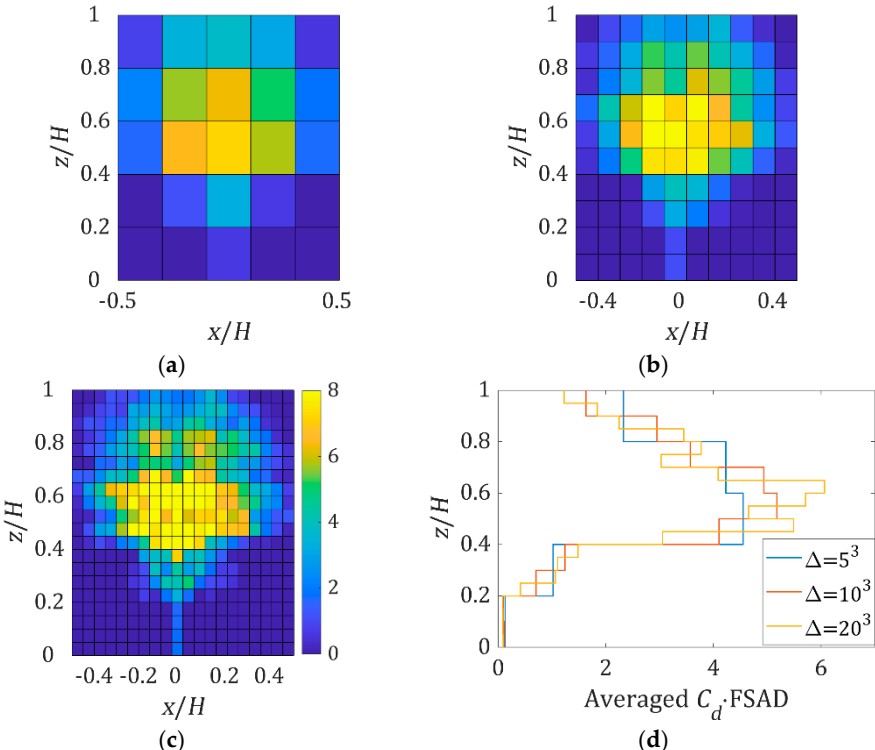

**Figure 8.** Averaged $C_d \cdot$ FSAD of *K. senegalensis* at 0° rotation $\left( \frac{\int C_d \cdot \text{FSAD} dy}{\Delta^{\frac{1}{3}}} \right)$: averaged along the streamwise direction (**a**) $\Delta = 5^3$; (**b**) $\Delta = 10^3$; (**c**) $\Delta = 20^3$; (**d**) further averaged along the *x*-direction $\left( \frac{\iint C_d \cdot \text{FSAD} dy dx}{\Delta^{\frac{2}{3}}} \right)$.

## 3. Results and Discussion

### *3.1. Velocity Profile*

An example of the center plane average flow field as captured by the PIV in the wind tunnel is shown in Figure 9. From the flow field, the velocity deficit can be clearly seen. We then extracted the velocity profile at 1*H* upstream, 1*H* downstream, and 2*H* downstream, for comparison with the simulation results.

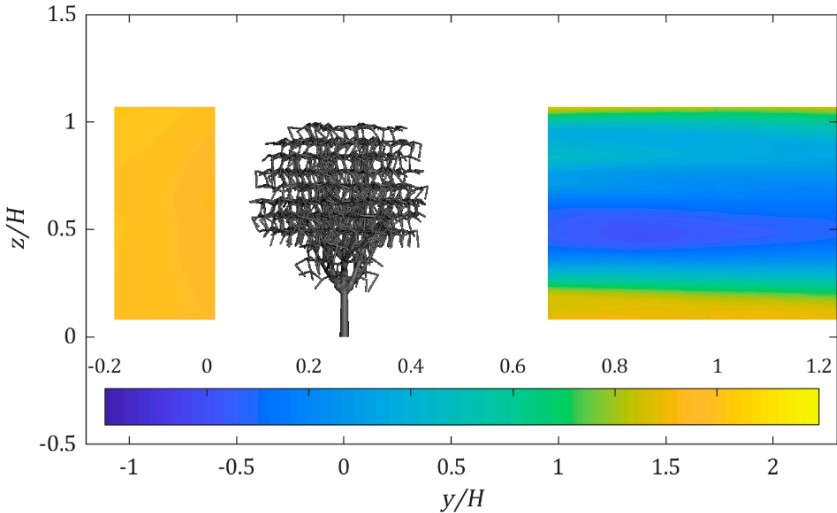

**Figure 9.** Normalized average horizontal velocity contour ($u_y/U_{\text{ref}}$) of *K. senegalensis* at $U_{\text{ref}}$ = 5 m/s and 0° rotation as measured from the wind tunnel experiment.

An example of the wake behind the tree models in the simulations is shown in Figure 10. The velocity contour at $Y = 0$ is plotted across the porous affected region. It shows the physical tree is absent in the domain, but its existence was being modelled and shows the velocity deficit. The wake flow develops and expands as the flow moves downstream. The fine structures diminish as the flow travels downstream until the wake contour becomes round and smooth.

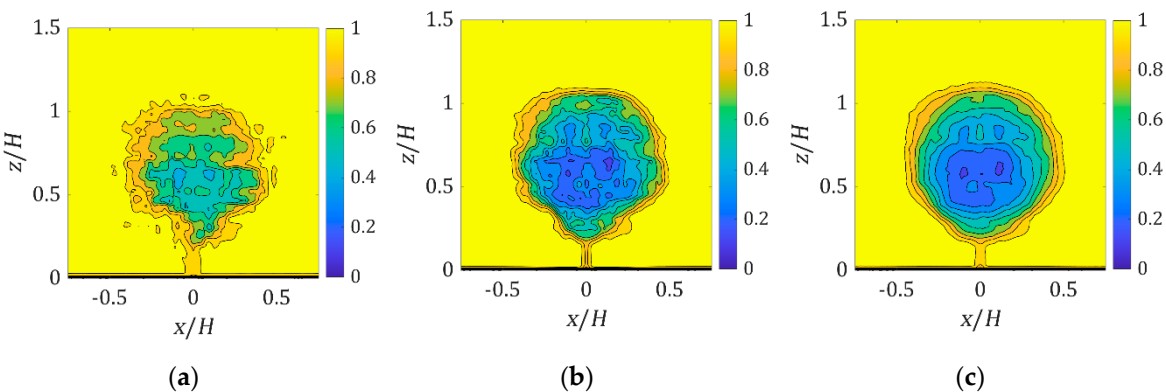

**Figure 10.** Simulated wake contours $\left(u_y/U_{\text{ref}}\right)$ behind the tree model at (**a**) $Y = 0H$; (**b**) $Y = 1H$; (**c**) $Y = 2H$.

Figure 11 shows that the average velocity profiles converge as the discretized grid resolution is increased. As the grid resolution increases, the finer flow features are revealed in the velocity profiles. However, the computational power required for $\Delta = 20^3$ increased by 2.5 times compared to $\Delta = 10^3$. As seen in the figure, the results at $\Delta = 10^3$ and $\Delta = 20^3$ are very close. We hence chose to simulate most of the cases at a resolution of $\Delta = 10^3$, except for *P. pterocarpum* and *S. saman*, which were simulated at $\Delta = 20^3$ due to the significantly wider crowns.

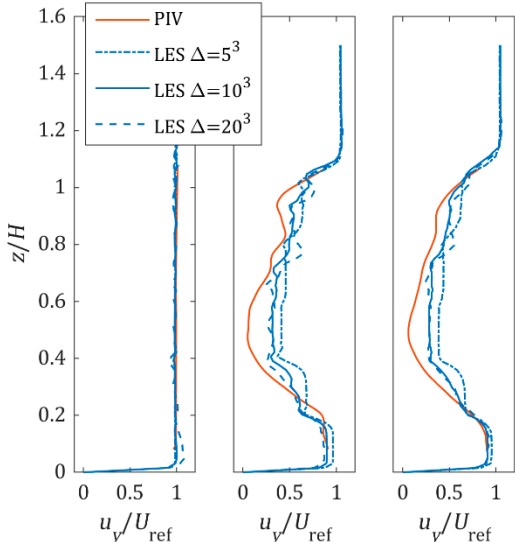

**Figure 11.** Average horizontal velocity profile at 1*H* upstream (**left**), 1*H* downstream (**middle**), and 2*H* downstream (**right**) of *K. senegalensis* 0° rotation, 15 m/s upstream wind speed.

Each model was simulated at 15 m/s wind speed and three rotational angles at 0°, 45°, and 90°. The velocity contour at 1*H* behind *K. senegalensis* is shown in Figure 12. The contours for other models are presented in Figure A8 in Appendix C. The asymmetric wake characteristic was captured for all the species simulated. In the velocity contours, the greatest velocity deficit is clearly shown behind the denser crown region. The coarse tree crown region and empty space have less impact on the velocity deficit. All the cases have distinguishable differences that can be related back to their own frontal views shown in Figure 2c and Figures A1–A6. *P. pterocarpum* and *S. saman* have a wider wake due to their wider tree crowns.

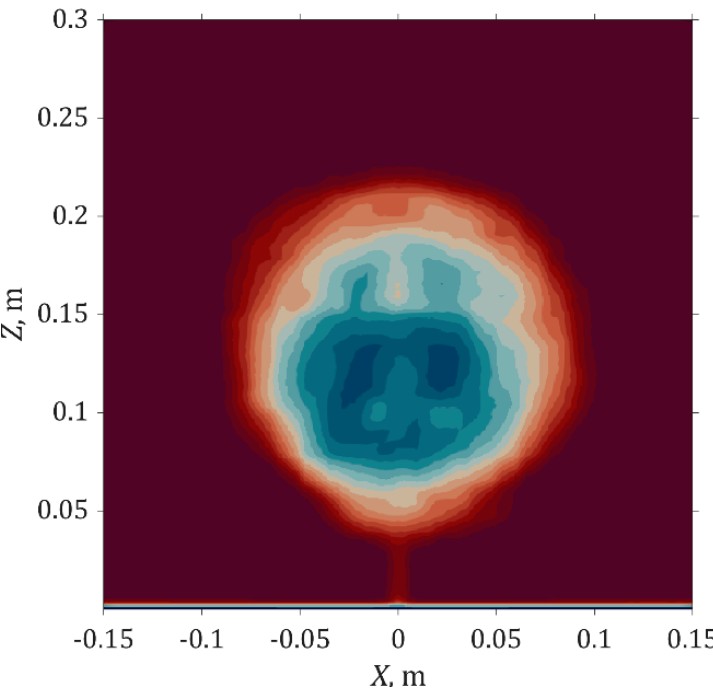

**Figure 12.** Simulated wake contour at 1*H* downstream of *K. senegalensis* for 0° rotation angle.

Figure 13 shows the comparison between the velocity profiles obtained from PIV measurements and LES simulations. It is observed that the velocity profiles measured at different wind speeds are almost identical. This implies that the test cases are insensitive to wind speed, similar to the observation on inflexible tree models in earlier studies [20,22]. Since the tests were all started at

random time, the similarity of the velocity profiles at different wind speeds also implies that the average velocity profiles are not significantly biased, despite the short duration of the PIV measurements.

Figure 13 also shows that the utilization of the discretized model managed to capture the wake profile variation with tree height. The simulation results are in reasonable agreement with the PIV measurements, except for *z/H* between 0.4 and 0.6, where the tree crowns are the densest. Generally, the simulations tend to underestimate the velocity deficit, and the simulated averaged velocity deficit error is between −19% and −22% compared to PIV measurements. Although it is considerably worse compared to the 7% for the summer tree reported by Dellwik et al. [11], it is actually comparable to the −20% for the bare winter tree reported in the same study. Considered that Dellwik et al. implemented a tuning process to obtain the best simulation results compared to the experiments, this could imply that −20% error is the limit of the current simulation schemes on inflexible trees.

Nonetheless, varying the local drag coefficient across the discretized volume should further improve the results, but the task to determine the local drag coefficients would be an enormous undertaking. It was also mentioned that the LES subgrid scale used in this study is the WALE subgrid scale model. The large flow features induced by the tree model are described in the momentum equations by the momentum sink. However, the subgrid scale model could be further improved by accommodating the tree modelling effects at a finer scale.

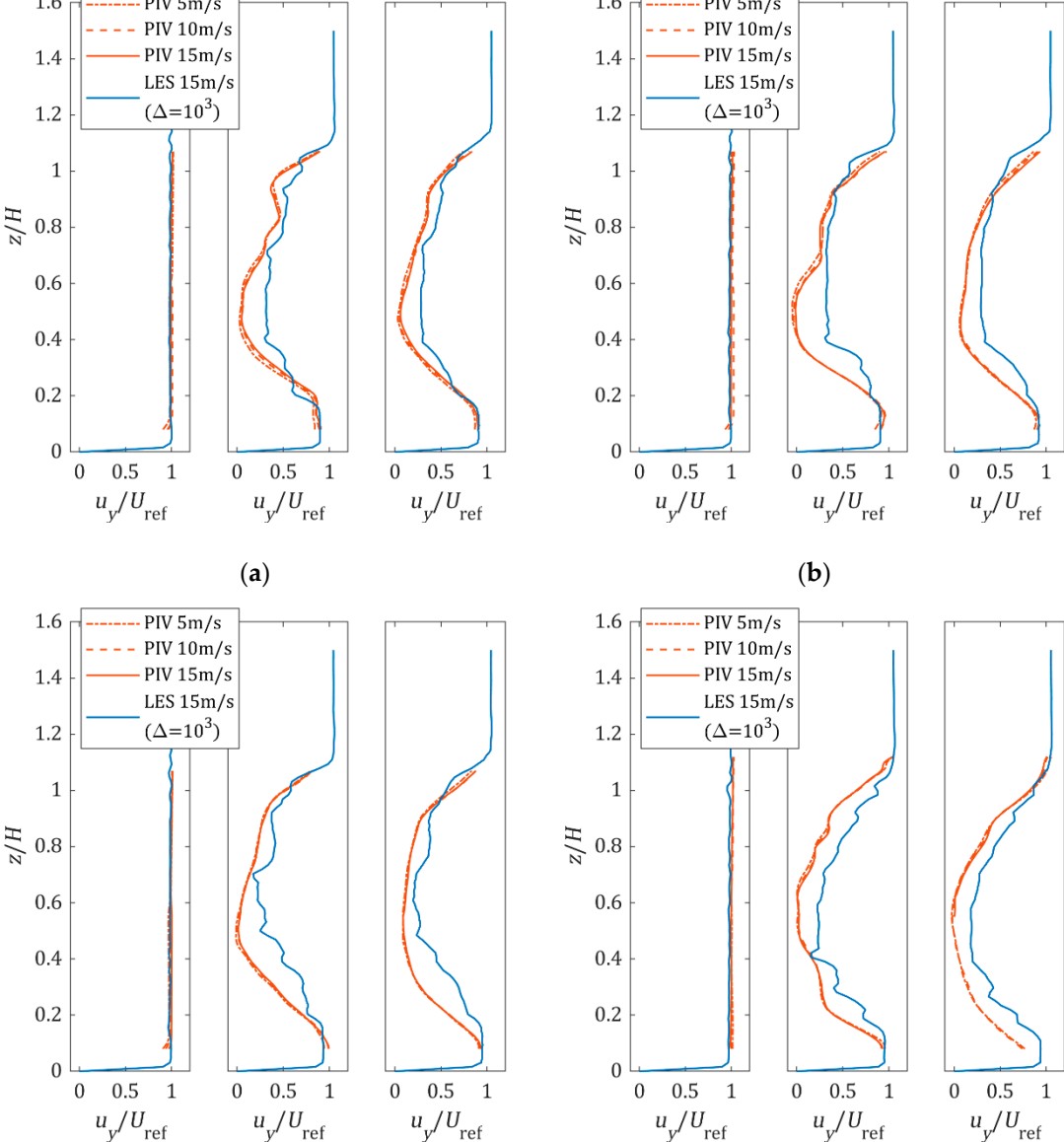

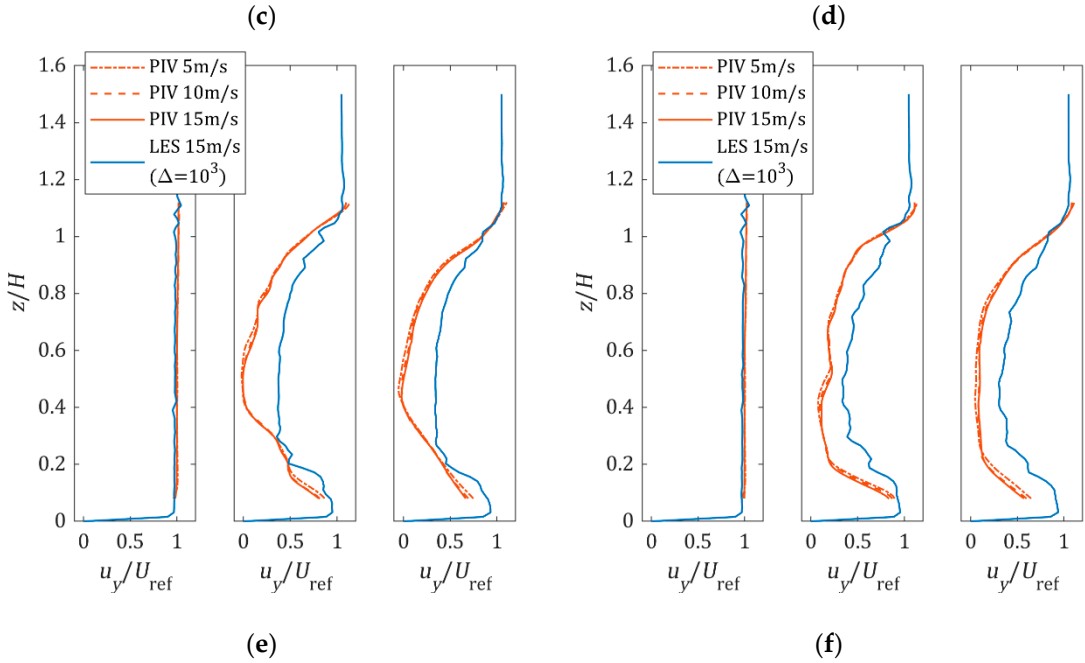

**Figure 13.** Average horizontal velocity profile at 1*H* upstream (**left**), 1*H* downstream (**middle**), and 2*H* downstream (**right**): (**a**) *K. senegalensis* 0° rotation; (**b**) *K. senegalensis* 45° rotation; (**c**) *K. senegalensis* 90° rotation; (**d**) *H. odorata* 0° rotation; (**e**) *H. odorata* 45° rotation; (**f**) *H. odorata* 90° rotation.

*3.2. Drag Force*

The total drag on the simulated tree model was collected throughout the run time. Time averaging was applied to the recorded drag and compared to the wind tunnel results. The time averaged drag on the simulated tree changes with the simulated tree model resolution. The drag difference between simulated tree and wind tunnel decreases as the tree model resolution becomes finer as shown in Table 2. The plus-minus sign indicates the standard deviation of the measurement.

**Table 2.** Drag comparison between simulation and experiment.

|  | *K. senegalensis* at 0° Rotation | | | |
| --- | --- | --- | --- | --- |
|  | Experiment | Simulations | | |
|  |  | $\Delta = 5^3$ | $\Delta = 10^3$ | $\Delta = 20^3$ |
| **Drag, N** | $2.040 \pm 0.016$ | 1.90 | 2.00 | 2.02 |
| **Difference, %** |  | −6.75 | −2.03 | −0.86 |

In Table 3, the drag on the tree models is compared with the experiment results. Almost all the simulations have less than ±5% difference except for *P. pterocarpum* and *S. saman*. This is mainly due to their larger crown volume compared to the other models. This methodology has consistent results across different tree models with reasonable error when compared to the experiments.

**Table 3.** Measured and simulated drag.

| Tree Species | Rotation Angle, ° | Drag, N | | Difference, % |
|---|---|---|---|---|
| | | Experiment | Simulations | |
| *K. senegalensis* | 0 | 2.040 ± 0.016 | 2.00 | −2.03 |
| | 45 | 2.022 ± 0.007 | 2.11 | 4.18 |
| | 90 | 1.896 ± 0.007 | 2.00 | 5.41 |
| *H. odorata* | 0 | 2.402 ± 0.011 | 2.35 | −2.00 |
| | 45 | 2.286 ± 0.016 | 2.28 | −0.08 |
| | 90 | 2.262 ± 0.013 | 2.27 | 0.23 |
| *S. saman*[1] | 0 | 3.116 ± 0.039 | 3.13 | 0.34 |
| | 45 | 3.080 ± 0.041 | 3.27 | 6.01 |
| | 90 | 2.745 ± 0.012 | 2.88 | 4.96 |
| *S. macrophylla* | 0 | 1.842 ± 0.025 | 1.87 | 1.43 |
| | 45 | 1.700 ± 0.011 | 1.79 | 5.17 |
| | 90 | 1.526 ± 0.009 | 1.54 | 1.12 |
| *S. grande* | 0 | 1.749 ± 0.016 | 1.72 | −1.56 |
| | 45 | 1.608 ± 0.013 | 1.64 | 1.77 |
| | 90 | 1.658 ± 0.014 | 1.72 | 3.94 |
| *T. rosea* | 0 | 1.969 ± 0.012 | 1.92 | −2.49 |
| | 45 | 1.584 ± 0.017 | 1.57 | −1.14 |
| | 90 | 1.056 ± 0.007 | 1.04 | −1.14 |
| *P. pterocarpum*[1] | 0 | 4.288 ± 0.022 | 4.65 | 8.33 |
| | 45 | 3.814 ± 0.044 | 4.09 | 7.20 |
| | 90 | 3.355 ± 0.016 | 3.74 | 3.93 |

[1] Simulated at $\Delta = 20^3$.

In this work, the total frontal silhouette area is proposed to be the reference area for the calculation of local drag coefficient used in the simulations. By doing so, we established the connection between the bulk drag and the local drag coefficient as seen in Equations (1) and (5). This method is an improvement over the practice of calibrating the local drag coefficient to fit the experiment result [11,12]. While the method works well in our study, it is still an open question as to whether it will work on a full-scale flexible tree.

*3.3. Bulk Drag Coefficients*

We further tested the tree models, measuring the drag from −90° to 90° rotation angle at every 5° interval. The result is 37 different orientations for each model at a particular wind speed. The bulk drag coefficients are plotted against the frontal optical porosity (*β*) in Figure 14. Included in the figure is data on flexible trees and models extracted from earlier study [22]. The frontal optical porosity of the flexible trees and models was calculated at rest. As seen in the figure, the spread of the drag coefficients of our tree models is narrower than the flexible trees and models. The spread of drag coefficient for natural trees can range from 0.25 to 1.25 [22].

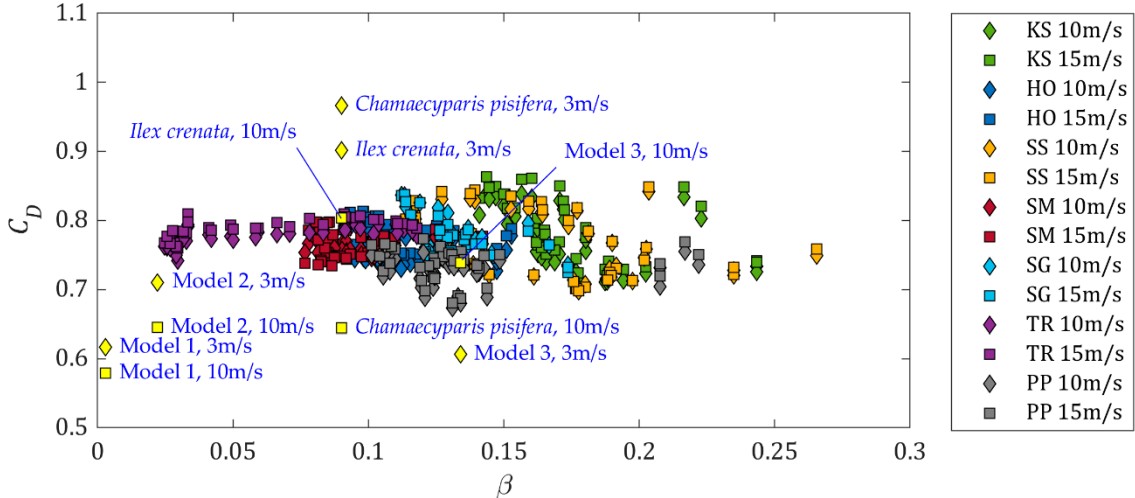

**Figure 14.** Bulk drag coefficient against frontal optical porosity of the tree models. Data of the specified flexible models/trees are data extracted from Manickathan et al. [22]. The reference area is the frontal crown area.

Table 4 summarizes bulk drag coefficients extracted from earlier studies [18,19,21,37]. The bulk drag coefficients measured in this study are between 0.674 and 0.863, which is within the range of the values shown in Table 4. However, it is worth noting that the tree models in this study were modelled based on mature trees. Hence, the comparison of the bulk drag coefficients with the saplings or small trees must be interpreted with reservations. Nonetheless, the bulk drag coefficients of the tree models are also comparable to that of real mature *S. grande* and *K. senegalensis* at low wind speeds, between 1 and 2 m/s.

**Table 4.** Bulk drag coefficients of real trees extracted from previous studies.

| Tree Species | Wind Speed, m/s | $C_D$ | Tree Species | Wind Speed, m/s | $C_D$ |
|---|---|---|---|---|---|
| *Populus trichocarpa* [1] | 5 | 0.780 | *Thuja plicata* [2] | 5 | 0.886 |
| | 10 | 0.642 | | 10 | 0.696 |
| | 15 | 0.574 | | 15 | 0.596 |
| *Populus tremuloides* [1] | 5 | 0.817 | *Hibiscus syriacus* [3] | 5 | 0.607 |
| | 10 | 0.688 | | 10 | 0.531 |
| | 15 | 0.647 | | 15 | 0.491 |
| *Alnus rubra* [1] | 5 | 0.738 | *Thuja occidentalis* [3] | 5 | 0.856 |
| | 10 | 0.595 | | 10 | 0.791 |
| | 15 | 0.551 | | 15 | 0.753 |
| *Betula papyrifera* [1] | 5 | 0.765 | *Ilex crenata* [3] | 5 | 0.807 |
| | 10 | 0.660 | | 10 | 0.780 |
| | 15 | 0.640 | | 15 | 0.765 |
| *Acer macrophyllum* [1] | 5 | 0.813 | *S. grande* [4] | 1 | 1.521 |
| | 10 | 0.635 | | 2 | 0.509 |
| | 15 | 0.599 | | 3 | 0.319 |
| *Tsuga heterophylla* [2] | 5 | 1.117 | *K. senegalensis* [4] | 1 | 1.565 |
| | 10 | 1.030 | | 2 | 0.506 |
| | 15 | 0.941 | | 3 | 0.390 |
| *Pinus contorta* [2] | 5 | 1.037 | | | |
| | 10 | 0.940 | | | |
| | 15 | 0.836 | | | |

[1] Hardwood saplings in wind tunnel. $C_D$ calculated using wind-speed-specific frontal area [18]. [2] Conifer saplings in wind tunnel. $C_D$ calculated using wind-speed-specific frontal area [19]. [3] Real

trees in wind tunnel. $C_D$ calculated using wind-speed-specific frontal area [21]. [4] Real mature trees in field tests. $C_D$ calculated using still air frontal area [37].

From Figure 14, the drag coefficients of *T. rosea* seem to increase mildly with the frontal optical porosity for optical porosity less than 0.09. After that, the coefficients for all tree models generally decrease with frontal optical porosity. Frontal optical porosity is known to be an important parameter in the study of wind load on a windbreak [3,38–41], and the drag coefficient was observed to increase with porosity, and peaked at aerodynamic porosity of 0.3 [22,39] before it starts to decrease. That value is called the critical aerodynamic porosity. As explained by Manickathan et al. [22], flow goes around the tree and recirculates behind the tree if the porosity is below the critical value. Above this critical value, airflow goes primarily through the porous tree crown. Aerodynamic porosity can be estimated using an empirical equation, show here as Equation (8) [39].

$$\alpha = \beta^{0.4} \tag{8}$$

$\alpha$ = Aerodynamic porosity;
$\beta$ = Frontal optical porosity.

Optical porosity of 0.09 is equivalent to aerodynamic porosity of 0.38, close to the 0.3 critical porosity reported. Figure 15 shows the bulk drag coefficients plotted against the estimated aerodynamic porosity. In order to show the general trend, the outliers were ignored. The outliers were selected visually since the bulk drag coefficients of each tree models form their own clusters, making the outliers obvious. Each cluster of data has a decreasing trend, except for *T. rosea* as this data cluster is spread across the critical region. The models are not made to be symmetrical. Hence, as the models rotate, both the porosity and shape change. The outliers of the data are most likely caused by the shape change.

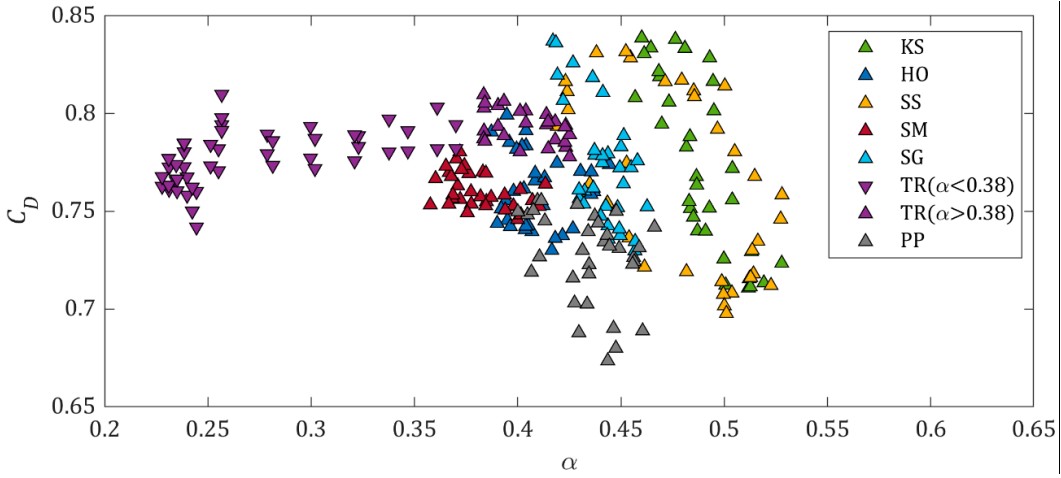

**Figure 15.** Bulk drag coefficient against aerodynamic porosity of the tree models.

Based on this observation, it is evident that the crown shape is also an important parameter that dictates the drag coefficient, and its effect could be more dominant than frontal optical porosity. The tree crowns of broadleaved trees are often complex, but they are less studied compared to conifers. Previous studies put emphasis on the dynamic response of the complex tree crown [42], but the effect on bulk drag coefficient was not explored. A systematic study on the effect of complex crown shape to bulk drag coefficient would undoubtedly contribute to better wind load estimation on broadleaved trees at different wind directions.

## 4. Conclusions

In this work, we studied the wind loading on seven major urban tree species in Singapore using scaled down fractal tree models. The species-specific 3D-printed tree models were constructed based

on measurements and statistical data taken on the real trees. Velocity profiles and drag were measured in the wind tunnel and the data were used to validate the results of LES simulations. We discretized the tree models within the simulations and represented them as momentum sinks. The results show that the current simulation scheme underestimates the velocity deficit but predicts the bulk drag at acceptable accuracy. Most importantly, we established a connection between the bulk drag and the local drag coefficient used in the simulations. This is an improvement over the calibrating process practiced in earlier studies. Using the established connection, the wake of a tree can be simulated if the bulk drag and the FSAD are known. This can potentially be used to assess the urban airflow and help to improve urban planning and urban tree management.

The bulk drag coefficients measured in the wind tunnel fall within a reasonable range, and were comparable to the drag coefficients of real trees at low wind speed. It is observed that a complex crown shape could affect the bulk drag coefficient more dominantly as compared to frontal optical porosity. It is also observed that each tree model forms its own cluster, suggesting that an empirical model for bulk drag estimation is possible if the crown can be generalized into various shapes. That would potentially speed up the risk assessment on tree failures due to wind load.

Notwithstanding, the observations made and methodology established in this study may not be generally applicable to all types of trees and models. Our tree models used in this study are inflexible models, and the models are not exact replicas of real trees due to fabrication constraints. The methodology should be further analyzed and scrutinized for the cases of flexible trees. Scalability of the methodology and crown shape generalization must also be addressed in the future.

**Author Contributions:** Conceptualization, H.J.P. and B.C.K.; methodology, W.-L.C., Y.E., H.J.P., Z.G., D.J.W., C.W.C.L., L.G., D.C.B. and D.L.; software, W.-L.C., Y.E., H.J. P., Z.G., C.W.C.L. and L.G.; validation, W.-L.C., Y.C., Y.E., H.J.P., Z.G., and D.J.W.; formal analysis, W.-L.C., Y.E. and Z.G.; investigation, W.-L.C., Y.E., Y.C. and Z.G.; resources, H.J.P. and B.C.K.; data curation, W.-L.C., Y.E., H.J.P., and Z.G.; writing—original draft preparation, W.-L.C. and Y.E.; writing—review and editing, B.C.K., D.J.W., D.C.B., W.-L.C., Y.E. and L.G.; visualization, W.-L.C. and Y.E.; supervision, H.J.P. and B.C.K.; project administration, H.J.P.; funding acquisition, H.J.P. and B.C.K. All authors have read and agreed to the published version of the manuscript.

**Funding:** This research and APC was funded by National Research Foundation of Singapore, grant number NRF2017VSG-AT3DCM001-029. This research is part of the Virtual Singapore Programme.

**Acknowledgments:** The author would like to thank interns, Ryan Ho Hin Tsang and Gu Xiaoke for their assistance in image processing for the determination of optical porosity and conducting the wind tunnel tests.

**Conflicts of Interest:** The authors declare no conflict of interest. The funders had no role in the design of the study; in the collection, analyses, or interpretation of data; in the writing of the manuscript, or in the decision to publish the results.

## Appendix A

Process of tree crown development for the tree models. The figures are not at the same scale.

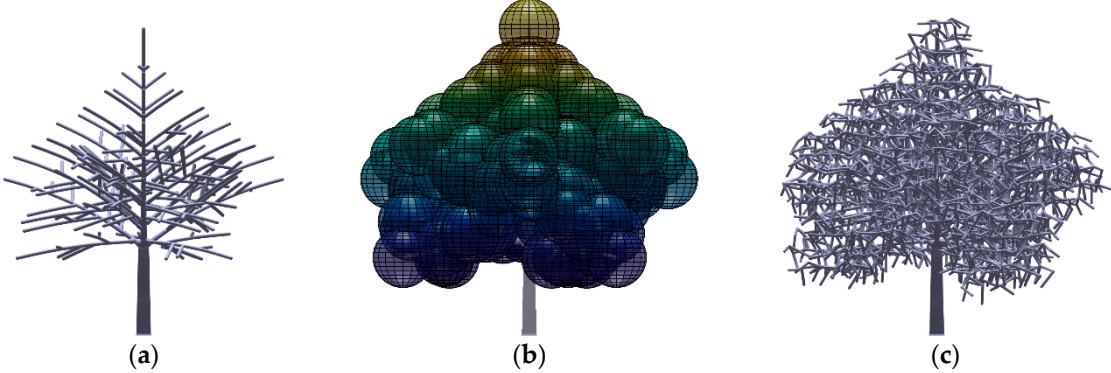

(a) (b) (c)

**Figure A1.** *H. odorata* fractal tree model: (**a**) trunk and branches; (**b**) tree crown volume; (**c**) final model (frontal view at 0° rotation).

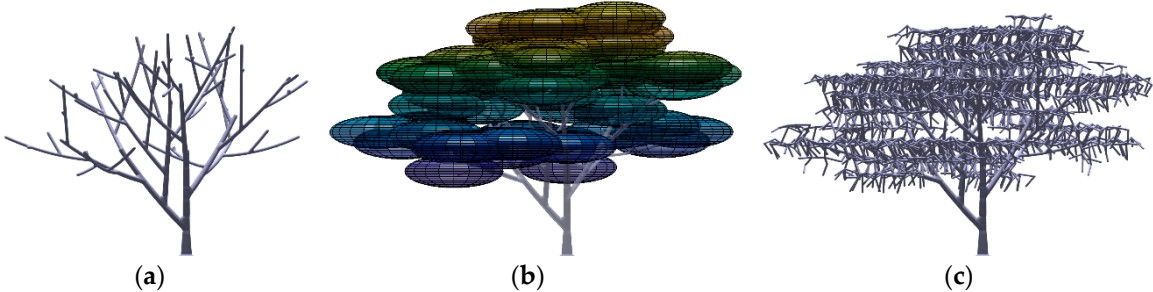

**Figure A2.** *S. saman* fractal tree model: (**a**) trunk and branches; (**b**) tree crown volume; (**c**) final model (frontal view at 0° rotation).

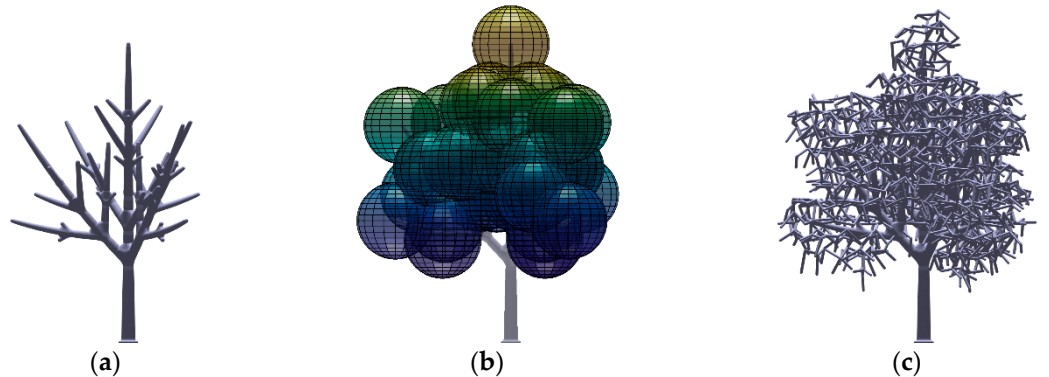

**Figure A3.** *S. grande* fractal tree model: (**a**) trunk and branches; (**b**) tree crown volume; (**c**) final model (frontal view at 0° rotation). .

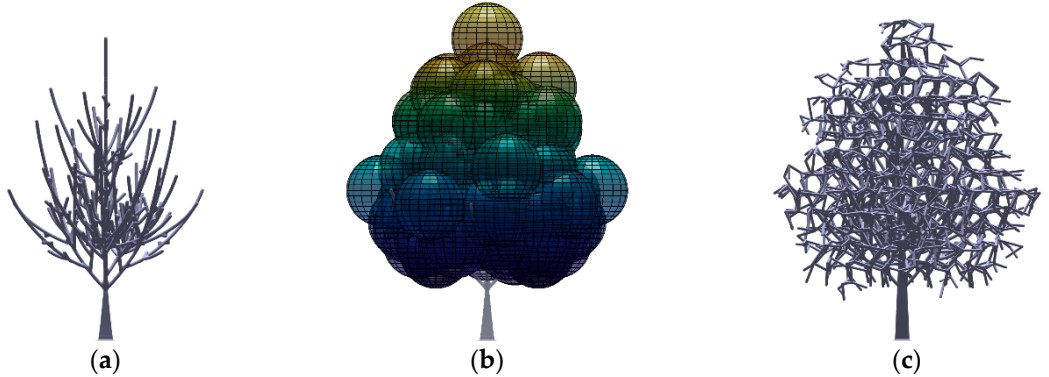

**Figure A4.** *S. macrophylla* fractal tree model: (**a**) trunk and branches; (**b**) tree crown volume; (**c**) final model (frontal view at 0° rotation).

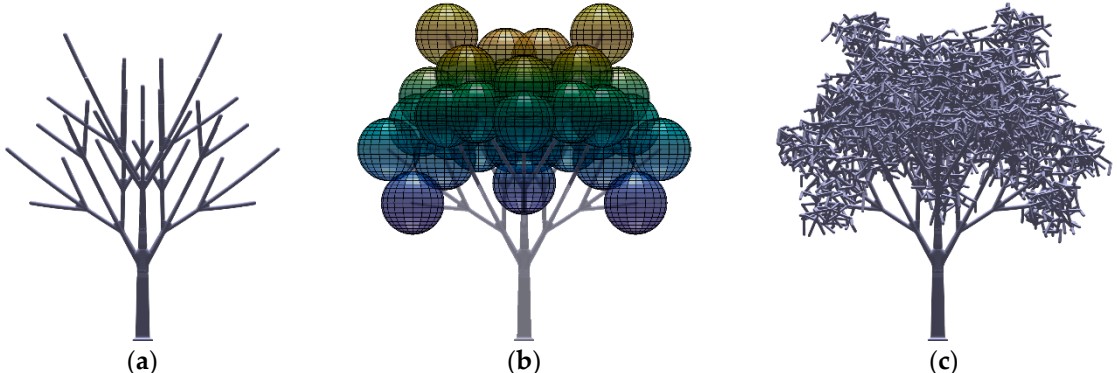

**Figure A5.** *T. rosea* fractal tree model: (**a**) trunk and branches; (**b**) tree crown volume; (**c**) final model (frontal view at 0° rotation).

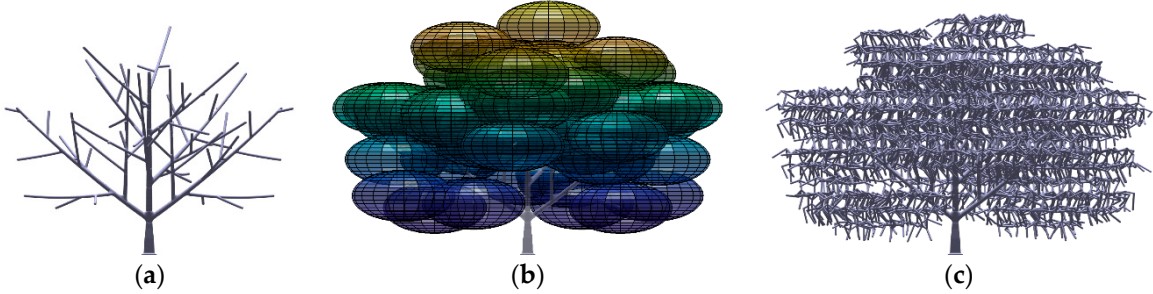

**Figure A6.** *P. pterocarpum* fractal tree model: (**a**) trunk and branches; (**b**) tree crown volume; (**c**) final model (frontal view at 0° rotation).

## Appendix B

Streamwise averaged $C_d \cdot$ FSAD for 0° rotation angle. The color map of the figures are at the same scale.

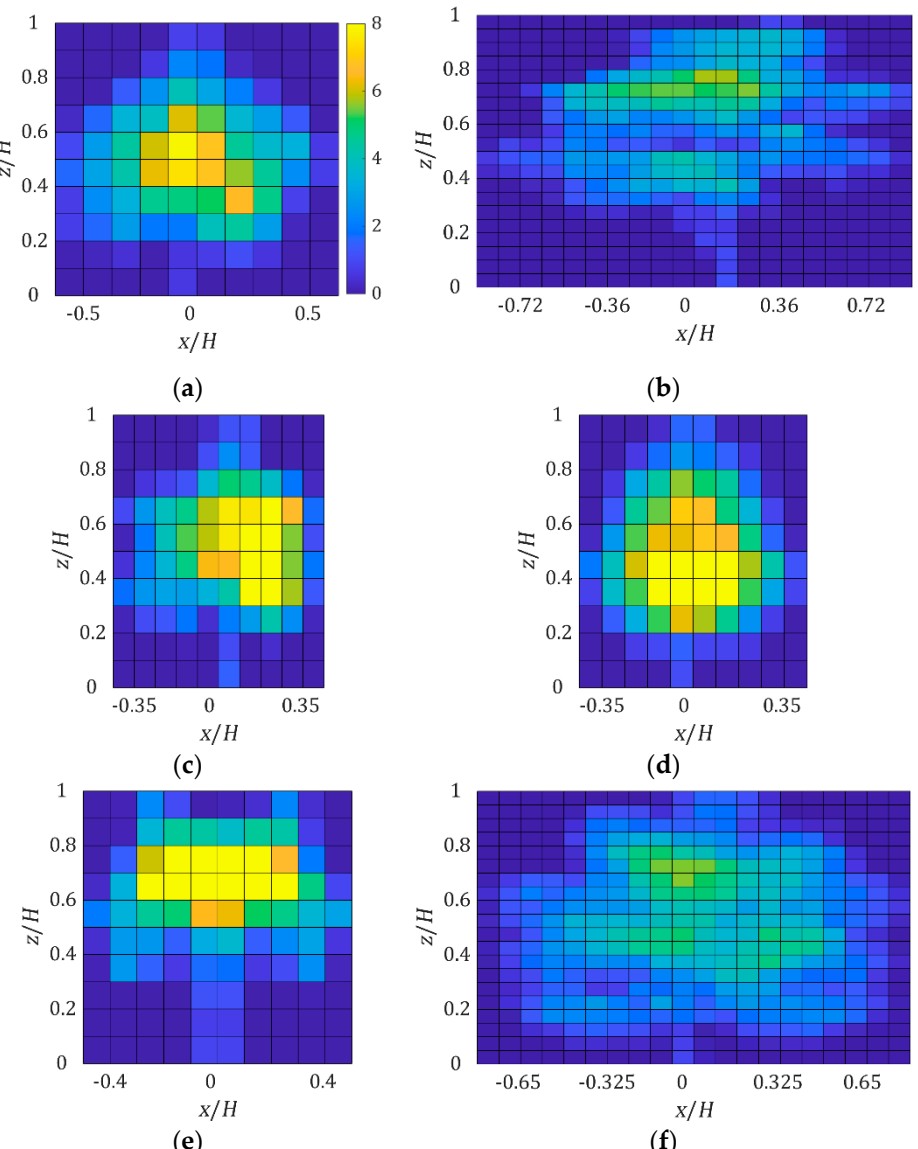

**Figure A7.** Streamwise averaged $C_d \cdot$ FSAD for 0° rotation angle: (**a**) *H. odorata* ($\Delta = 10^3$); (**b**) *S. saman* ($\Delta = 20^3$); (**c**) *S. grande* ($\Delta = 10^3$); (**d**) *S. macrophylla* ($\Delta = 10^3$); (**e**) *T. rosea* ($\Delta = 10^3$); (**f**) *P. pterocarpum* ($\Delta = 20^3$).

## Appendix C

Wake contour at 1*H* downstream.

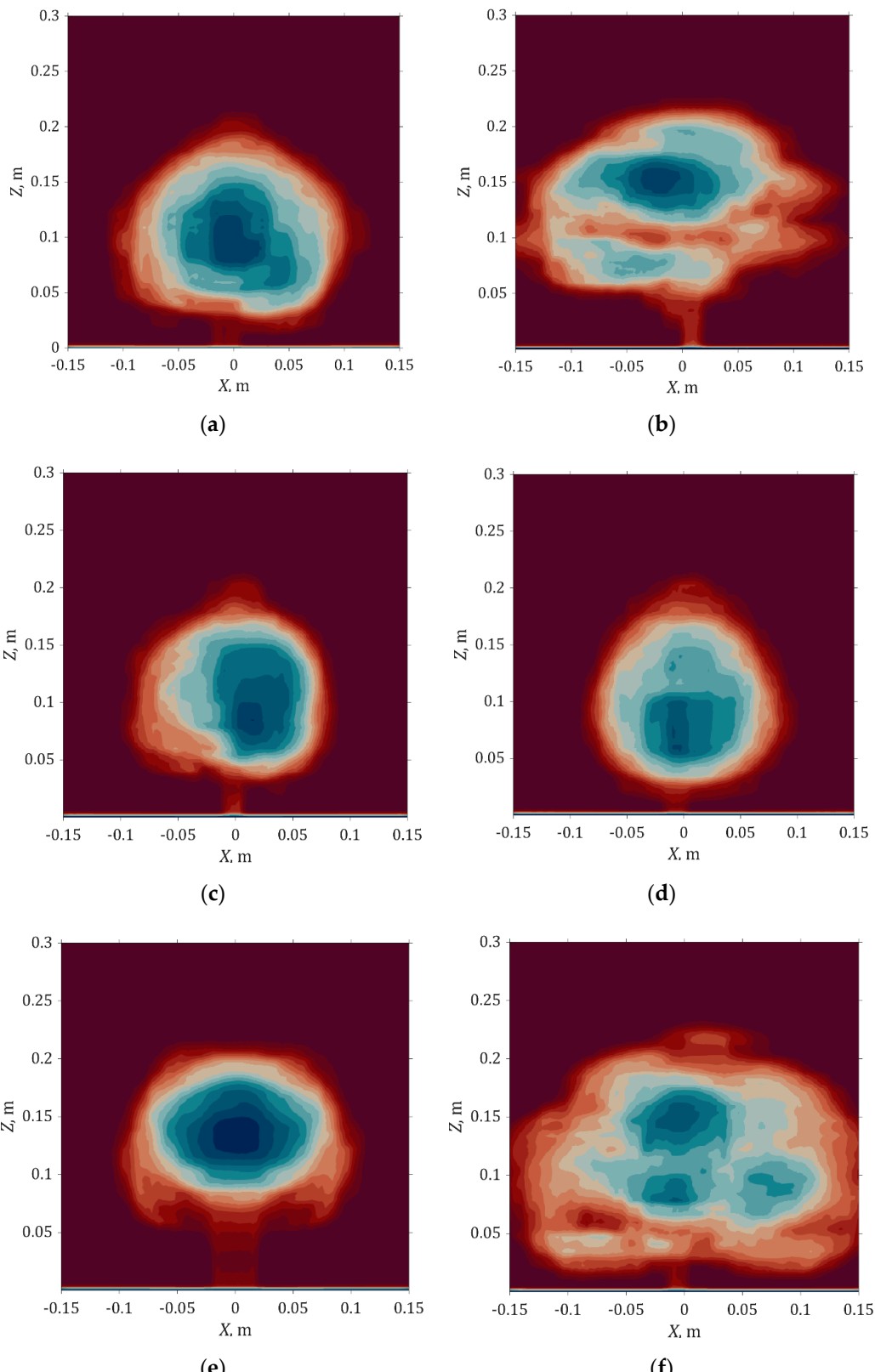

**Figure A8.** Wake contour at 1*H* downstream for 0° rotation angle: (**a**) *H. odorata*; (**b**) *S. saman*; (**c**) *S. grande*; (**d**) *S. macrophylla*; (**e**) *T. rosea*; (**f**) *P. pterocarpum*.

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
