# Peer review of "Wind Loading on Scaled Down Fractal Tree Models of Major Urban Tree Species in Singapore"

_forests, doi:10.3390/f11080803_

Round 1
Reviewer 1 Report
The manuscript deals with the connection between the bulk drag and the local drag from trees for use in numerical simulations, which is a very important in the parameterization of the way that tress are accounted for in numerical simulations. The topic is very relevant for e.g. wind energy, park design and forestry. The manuscript constitute a novel and important contribution to the ongoing work how to parameterize the effect of trees in numerical simulations.
Some minor comments:
1. Line 79: is it “we chose to conduct experiments” or “we conducted experiments”. Please clarify.
2. Line 108: Explain in more detail what is meant by “modelling based on species-specific growth processes” and “L-systems”, and why it is important. I had the impression that the trees mainly were modelled based on laser scanning of real trees.
3. Line 179-180: Be more explicit about the difference between U and ui; and why it is not ui-squared instead of ui times U.
4. In Eq. (1) Cd is the bulk drag coefficient, in line 206 it is the local drag coefficient –please distinguish between the two symbols.
5. Line 270. Why is there a dependence on the flow direction. I would think that the tress are symmetric? This need clarification
Reviewer 2 Report
Paper is a really interesting one in that it links model trees made using a novel technique (3D printing), wind tunnel measurements and LES simulations.
I would like to see this published but it requires some work to be acceptable. None of the work is substantial, which is why I have suggested that this is minor revision.
- Somebody needs to read through the text and check the English. It is mainly fine but occasionally becomes confusing. I have made a number of suggested changes in the attached annotated manuscript.
- Line 65. I think suggesting the discretized momentum sink was used to "simulate the flow" is not quite correct. I would prefer "modify the flow" or "simulate the tree drag" would be better.
- Line 109. I think you need to explain what is a L-system to normal readers of Forests.
- Line 116. "Thickened" by how much?
- Section 2.1. A key issue that you do not discuss is the lack of Reynolds number scaling. You cannot get correctly scaling because of the small size of your models. Therefore, drag will be not predominately pressure/form drag as it would be at full scale but a much larger proportion of the drag will be viscous drag at the surface of the model canopy. You need to discuss this even if it is unavoidable.
- Line 136. What accuracy do you estimate ±%?
- Line 139. What else would you take into account other than the "outer boundary of the leaves"? Do you take account of the stem also? I don't really understand what you are trying to say here.
- Table 1. Normally porosity is given as a %.
- Line 152-153. Please give manufacturer's name and city for all equipment. Also it will not be clear to many readers that PIV makes the wind speed measurements.
- Line 155. Can you give a web link or reference for the DaVis software.
- Figure 4. There are no dimensions on the figure so we know the width and height of the tunnel.
- Line 159. The load cell is not totally clear in your diagram. Is this a force or torque measurement? I presume force.
- Section 2.2. There is not enough information on the wind tunnel. where is the fan relative to the model, are there flow straighteners etc? In particular how long and wide is the working section. I presume this is not boundary-layer flow but closed-channel (pipe) flow with roughly constant wind speed across the tunnel (rapid decrease at sides). This is what Fig. 10 indicates.
- Line 166. Is air density adjusted for the temperature in the tunnel?
- Line 172. What do you mean by "promising" and please provide a reference.
- Figure 5. Again no dimensions.
- Line 194. "Zero gradient" in what? Do you assume zero gradient in streamwise velocity?
- Line 194: What does a convergence criteria of 10-6 actually mean? And how did you define statistically stable? To what limit?
- Lines 206 and 207. These terms do not appear in equations 6 and 7.
- Line 217. Or rather it is the area "represented by each pixel".
- Lines 225 and 226. What do you mean by Δ=53, Δ=103 and Δ=203? Not at all clear.
- Figure 7. I became confused between streamwise and longitudinal. Are (a), (b) and (c) across the wind tunnel? This would make it the average Cd.FSAD in the across flow plane (x-z plane) and not streamwise. And is (d) the accumulated value down the tunnel?
- Section 3.1. You need to be clear that this is from the wind tunnel.
- Figure 9. A little difficult to follow. Could you just plot each of the contour plots in 2D separately and tell the reader where the plot is from. At the moment there is no scaling on the figure (it has to be in tree heights but this is not clear on the figure).
- Figure 10. I wonder if different colours would be better rather than all LES results in blue.
- Line 242-243. I think this is for Materials and Methods.
- Lines 263-264. Why did you run at different wind speeds? Did you expect to find differences?
- Figure 12. You make no effort to estimate the difference between PIV and LES. i think you need to do that. The LES seems to miss the severe wind speed reductions between z/H = 0.4 to 0.6.
- Line 283. Why do you discuss fluctuating drag? This is not a realistic atmospheric boundary-layer so I don't think the fluctuating loads are particularly meaningful.
- Table 2. Can you provide some uncertainty to these measurements? You at least measured the wind tunnel values 3 times.
- Table 3. Did you work out what the drag was if you just used a solid tree (zero porosity)? You sort of hint at it. i just wonder if your models could be much simplified by just creating the outer shell of the canopy.
- Line 300. What do you mean by "bulk drag coefficient"?
- Line 311. I really think when discussing drag coefficients of trees you need to reference Mayhead (1973), Rudnicki et al. (2004) and Vollsinger et al. (2005).
- Figure 14. The drag coefficient units (y-axis) make no sense. It is a dimensionless number. What are lower and upper dashed lines related to K. senegalensis? Not explained anywhere.
- Figure 15 and equations 9-15. This is the weakest part of the paper. I cannot see the value of this analysis. Most of the fits have very low R2 values. You could easily draw a straight line through all the data for all trees. There is possibly a very slight peak at around 0.35-0.4 but in fact the most interesting fact is that the aerodynamic porosity makes very little difference. Does this suggest that the exact details of the branch architecture of your trees is unimportant and it is the canopy cross-sectional area that is important? I would leave out all the equations and maybe only fit a polynomial of 2nd order to all your data.
- Conclusions. There is no Discussion. You need a Discussion. Maybe you can combine with Conclusions (Discussion and Conclusions). There needs to be discussion of your results against other work in the field, wind tunnel and computer simulations. For example, how does this work compare with Dellwik et al. (2019)? You might also add a list of potential future work. In addition how might your work be used? Who will use this wiork? Can it be used in urban climatology, urban flow modelling, risk management, etc. The context of the work is discussed at the beginning but it is not returned to at the end.
- So my main issue with the paper is that the work is not pulled together at the end and set in the context of other work and the value of this work for other researchers and practitioners.
- Line 355. I find debatable the point that the finer the discretization the better the fit. It seemed that once you had if Δ=103 there was no improvement at higher resolutions.
- Appendix A, Yes, very good.
